# Bone Disease in Nephropathic Cystinosis: Beyond Renal Osteodystrophy

**DOI:** 10.3390/ijms21093109

**Published:** 2020-04-28

**Authors:** Irma Machuca-Gayet, Thomas Quinaux, Aurélia Bertholet-Thomas, Ségolène Gaillard, Débora Claramunt-Taberner, Cécile Acquaviva-Bourdain, Justine Bacchetta

**Affiliations:** 1Pathophysiology, Diagnosis and Treatment of Bone Diseases, INSERM UMR 1033, 69008 Lyon, France; irma.machuca-gayet@inserm.fr (I.M.-G.); itommy640@gmail.com (T.Q.); declata@gmail.com (D.C.-T.); 2Centre de Référence des Maladies Rénales Rares, Centre de Référence des Maladies Rares du Calcium et du Phosphore, Hôpital Femme Mère Enfant, 69500 Bron, France; aurelia.bertholet-thomas@chu-lyon.fr; 3INSERM CIC 1407, CNRS UMR 5558 and Service de Pharmacotoxicologie Clinique, Hospices Civils de Lyon, 69500 Bron, France; Segolene.Gaillard@chu-lyon.fr; 4Service de Biochimie et Biologie Moléculaire, Groupement Hospitalier Est, 69500 Bron, France; cecile.acquaviva-bourdain@chu-lyon.fr; 5Faculté de Médecine Lyon Est, Université de Lyon, 69008 Lyon, France

**Keywords:** Renal osteodystrophy (ROD), CKD-MBD, Orphan disease, Bone, Nephropathic cystinosis, mTor signaling, Osteoblast, Osteoclast

## Abstract

Patients with chronic kidney disease (CKD) display significant mineral and bone disorders (CKD-MBD) that induce significant cardiovascular, growth and bone comorbidities. Nephropathic cystinosis is an inherited metabolic disorder caused by the lysosomal accumulation of cystine due to mutations in the *CTNS* gene encoding cystinosin, and leads to end-stage renal disease within the second decade. The cornerstone of management relies on cysteamine therapy to decrease lysosomal cystine accumulation in target organs. However, despite cysteamine therapy, patients display severe bone symptoms, and the concept of “cystinosis metabolic bone disease” is currently emerging. Even though its exact pathophysiology remains unclear, at least five distinct but complementary entities can explain bone impairment in addition to CKD-MBD: long-term consequences of renal Fanconi syndrome, malnutrition and copper deficiency, hormonal disturbances, myopathy, and intrinsic/iatrogenic bone defects. Direct effects of both *CTNS* mutation and cysteamine on osteoblasts and osteoclasts are described. Thus, the main objective of this manuscript is not only to provide a clinical update on bone disease in cystinosis, but also to summarize the current experimental evidence demonstrating a functional impairment of bone cells in this disease and to discuss new working hypotheses that deserve future research in the field.

## 1. Introduction

Pediatric patients with chronic kidney disease (CKD) display significant abnormalities in bone and mineral metabolism, also known as CKD-MBD, for mineral and bone disorders associated with CKD. The pathophysiology of CKD-MBD is complex and multifactorial, multiple factors were identified such as abnormalities in calcium and phosphate metabolism, resistance to growth hormone (GH), modifications of the GH-insulin such as growth factor type 1 (IGF1) axis, hypogonadism, malnutrition, and drug toxicity (corticosteroids) [1]. Not only do these complications impact overall quality of life through their effects on both physical and mental well-being in children with CKD, but alterations in mineral metabolism and bone disease contribute to a significant decrease in life expectancy. Renal osteodystrophy (ROD) is considered to be part of the systemic CKD-MBD, characterized by one or a combination of the following abnormalities [2,3]: 1) abnormalities of calcium, phosphorus, PTH or vitamin D metabolism, 2) abnormalities in bone histology, linear growth or strength, and 3) vascular or other soft tissue calcification. The bone and growth long-term consequences of CKD were highlighted in a cohort of 249 young Dutch adults with onset of end-stage renal failure before the age of 14 years: in this cohort, 61% of patients had severe growth retardation, 37% severe bone disease (as defined by at least one of the following conditions: deforming bone abnormalities, chronic pain related to the skeletal system, disabling bone abnormalities, aseptic bone necrosis and low-traumatic fractures) and 18% disabilities resulting from bone impairment [4]. More recently, a significantly increased risk of fractures was demonstrated in the pediatric North American CKiD cohort, evaluating 537 children with CKD at a median age at inclusion of 11 years. At baseline, 16% of them had a history of fractures, and after a median follow-up of 3.9 years, 43 boys and 24 girls experienced fractures, corresponding to a fracture risk 2- to 3-fold higher than in general populations [5]. After transplantation, CKD-MBD may persist, and is mainly due to the preexisting renal osteodystrophy and cardiovascular changes at the time of transplantation, but also to corticosteroids and reduced graft function [6].

Moreover, in addition to “conventional” ROD and CKD-MBD, there is now evidence that some genetic renal diseases may also have a direct negative impact on bone per se, notably in primary hyperoxaluria, nephropathic cystinosis, autosomal dominant polycystic kidney disease, and Pierson syndrome [7,8,9,10]. The purpose of this review is to report the clinical update on bone pathology in nephropathic cystinosis and provide a summary of basic research findings on bone cell impairment that contribute to unbalanced bone remodeling in this orphan disease and finally to discuss new working hypotheses that deserve future research in the field.

### 1.1. The Emerging Concept of Cystinosis Metabolic Bone Disease

Nephropathic cystinosis is a rare autosomal recessive lysosomal storage disease caused by the lysosomal accumulation of cystine because of mutations in the *CTNS* gene encoding cystinosin, a 367 amino acids lysosomal cystine transporter with seven transmembrane domains. To date, over 140 pathogenic mutations have been reported. The most frequent one is a large deletion of 57 kb involving the first nine exons of the *CTNS* gene and part of exon 10, detected in approximately 50% of mutant alleles of patients of north European and North American origin. Other mutations include deletions, small insertions, duplications, missense, nonsense, splice site and promoter sequence mutations as well as genomic rearrangements. Severe truncating mutations affected both alleles are generally associated with infantile nephropathic cystinosis, the most frequent and severe form of the disease [11]. Cystinosis (1 to 9 / 100 000 persons) accounts for 1.4% of children on dialysis and 2.1% of pediatric renal transplant recipients in the USA [12].

Cysteamine therapy is a mainstay of its management, decreasing lysosomal cystine accumulation in target organs [8]. The use of cysteamine therapy since the 1980’s has postponed end-stage renal disease and extra-renal morbidities to the second or third decade of life [13]. Later on, the mechanisms of action were elucidated: cysteamine enters the lysosome and breaks cystine disulfide bond, resulting in the formation of cysteine and cysteamine-cysteine mixed disulfide. These can then leave the lysosome via the cysteine transporter and PQLC2, respectively. Cysteamine therefore restores the efflux of cystine by reacting with it, products of the reaction being able to bypass the deficient transporter [14,15]. However, as patients receive cysteamine and as global survival improves [8], bone impairment occurring during teenage or early adulthood was recently described as a “novel” complication of cystinosis [16,17,18,19,20], further leading to the concept of “cystinosis metabolic bone disease” (CMBD) [21]. Cysteamine almost doubled the time during which patients present with a significant renal phosphate leak due to the Fanconi syndrome; even though patients receive phosphate supplementation and sometimes active vitamin D analogs, this fact may explain at least partly, the current observed bone phenotype.

Indeed, this concept of CMBD is currently emerging, with the description in vivo of bone fragility and symptoms [18,19], and in vitro of a functional deficit both in osteoblasts and in osteoclasts [22,23]. Even though its exact underlying pathophysiology remains unclear, at least five distinct but complementary entities can explain CMBD in addition CKD-MBD and post-transplant CKD-MBD [21]: 1) long-term consequences of hypophosphatemic rickets and renal Fanconi syndrome (i.e., hypophosphatemia, metabolic acidosis, chronic hypovolemia, hypocalcemia and 1–25 OH_2_ vitamin D deficiency) together with iatrogenic effects of its supportive management, 2) deficiency in nutrition and micro-nutrition, and notably copper deficiency [24], 3) hormonal disturbances such as hypothyroidism, hypogonadism and hypoparathyroidism usually appearing in late teenage or early adulthood, and resistance to growth hormone and IGF1, 4) myopathy, and 5) intrinsic and iatrogenic bone lesions such as direct bone effects of *CTNS* mutation and cysteamine on osteoblasts and osteoclasts [21].

### 1.2. Bone Impairment in Patients Nephropathic Cystinosis: Clinical Facts

In animal models, the invalidation of the *CTNS* gene in certain strains of mice is associated neither with renal phosphate wasting nor with renal failure, but causes severe growth retardation and osteopenia with decreased mineralization and cortical thickness [25], thus raising the hypothesis of a specific underlying bone defect in cystinosis. Before 2016, there was little data in the literature describing bone impairment in patients with cystinosis [17,26,27]; one case of co-occurrence of *Osteogenesis Imperfecta* type VI and cystinosis as a contiguous gene syndrome had also been described [28].

We first reported in 2016 three cases of teenagers with cystinosis displaying a severe bone phenotype who underwent bone biopsies at the time of orthopedic surgery [20]: even though quantitative histomorphometry was not performed, all patients displayed increased bone resorption areas. The biopsy of the first patient showed normal mineralization with active areas for both bone resorption and formation, the biopsy of the second patient demonstrated active areas of mineralization and bone formation, with extended and unusual areas of bone resorption full of multi-nucleated osteoclats, and the biopsy of the third patient found diffuse defective mineralization with enlarged active osteoclasts but little osteoblastic activity. All these data, even preliminary and scarce, were in favor of increased resorption in these patients with bone impairment and cystinosis [20], and cannot be explained only by the underlying renal osteodystrophy induced by CKD. One year later, a review focused on specific bone symptoms in cystinosis, warning physicians to be cautious on this specific complication that may alter the quality of life of patients [16].

We therefore performed a pilot observational study in 10 patients (two receiving conservative therapies, two undergoing hemodialysis, and six with a past of renal transplantation) using biomarkers (total Alkaline Phosphatase ALP, parathyroid hormone PTH, 25 OH-vitamin D 25-D, and Fibroblast Growth Factor FGF23), Dual X-ray absorptiometry (DX at the spine and total body and High Resolution peripheral Quantitative Computed Tomography (HR-pQCT) at the ultra-distal tibia [18]. At a median age of 23 (range 10–35) years, seven out of ten patients (70%) complained of a bone symptom (past of fracture, bone deformations, and/or bone pain). Using HR-pQCT, significant decreases in cortical parameters and total bone mineral density were observed in cystinotic patients in comparison with controls at the tibia. There were no differences for trabecular parameters. Interestingly, there were no obvious abnormalities of circulating and urinary calcium as well as ALP levels, but biochemical tests showed a tendency toward low PTH and low FGF23 levels, likely reflecting chronic phosphate wasting because of low phosphate reabsorption rates, although there was no obvious uncontrolled Fanconi syndrome in patients with conservative management and in those after transplantation. When comparing patients with a functioning renal graft, spine Z-scores were significantly greater in patients with a past of nephrectomy in comparison to patients without nephrectomy of native kidneys. We concluded that chronic phosphate wasting may silently impair bone quality in cystinosis, since nephropathic cystinosis is a unique state of long-term CKD-MBD and renal phosphate wasting due to the Fanconi syndrome that persists even at late CKD stages. These preliminary data still need to be confirmed in larger cohorts, since this would provide a strong rationale for proposing nephrectomy of native kidneys to patients in order to decrease renal phosphate wasting and therefore improve bone quality. In conclusion, this pilot study confirmed for the first time that bone disease was clinically significant in teenagers and young adults with cystinosis, and that bone impairment was much more frequent in these patients than in patients with CKD for other causes [5,18]. Moreover, HR-pQCT allowed a better definition of this CMBD, impacting rather the cortical than the trabecular compartment, similarly to what had been published for the global knockout mice *Ctns*
^-/-^ [25].

In the meantime, a North American team performed bone and mineral evaluations of 30 patients (60% undergoing conservative management and 40% after renal transplantation) with cystinosis at a mean age of 20 (range 5–44) years, including history and physical examination, biochemicals, DXA (femoral neck, 1/3 radius, total hip), vertebral fracture assessment, skeletal radiographs, and renal ultrasounds [19]. Histomorphometric data were also reported in six additional subjects from another center. Mean bone mineral density (BMD) Z-scores decreased at all sites, and low bone mass at one or more sites was present in 46% of subjects. A significant proportion of patients also complained of bone symptoms: 27% reported one or more fractures of the long bones, 32% displayed incidental vertebral fractures, 64% suffered from deformities/bowing of the long bones and 50% had scoliosis. In contrast to our data, most patients presented with hypercalciuria, at least at the pre-transplant clinics that may also worsen bone lesions. Histomorphometric analyses showed impaired mineralization in four out of six patients, with no defined pattern in turnover or volume, and no mention of a specific osteoclastic trait. It is noteworthy that three bone biopsies were performed before the availability of cysteamine therapy, likely inducing a strong bias to interpret these results, since the management of “conventional” ROD has also improved over the last three decades [19]. Anyway, these authors also concluded that skeletal deformities, decreased bone mass, and vertebral fractures are common and relevant complications of nephropathic cystinosis, even before renal transplantation.

With these data in mind, international guidelines were published in 2019 to guide the diagnosis of CMBD and clinical management of these specific patients [21]. Briefly, assessment of CMBD involves following growth parameters, monitoring blood levels of phosphate, bicarbonate, calcium, and ALP, periodically obtaining bone radiographs, determining levels of critical hormones and vitamins, such as thyroid hormone, PTH, 25-D, and testosterone in males, and surveillance for non-renal complications of cystinosis such as myopathy. The multi-disciplinary and global management of CMBD includes replacement of urinary losses (i.e., mainly phosphate supplementation and alkalinization), cystine depletion with oral cysteamine, native vitamin D supplementation, hormone replacement (and notably recombinant growth hormone therapy early during the course of CKD) [29], physical therapy, and corrective orthopedic surgery [21].

### 1.3. Bone Disease and Nephropathic Cystinosis: A Functional Impairment of Both Osteoblasts and Osteoclasts

The study of bone phenotype in one month-old *Ctns* knockout (*Ctns^-/-^*) mice (of C57BL/6 background) reveals a decrease in bone mineral density (BMD), trabecular volume (TV) and number/thickness of bone trabeculae compared to control animals. Incidentally, these mice did not display any sign of Fanconi syndrome. Therefore these changes are not secondary to altered kidney functions and are associated with a reduction in the number of osteoclasts and osteoblasts within bone tissue in 1 month-old mice. *Ctns*
^-/-^ mice show a decrease in plasma markers of bone turnover, and notably procollagen type 1 amino-terminal propeptide and tartrate-resistant acid phosphatase [30].

Bone remodeling is a tightly regulated process involving bone resorption of old bone and bone formation to maintain a healthy skeleton. These two activities are coupled, through a complex crosstalk between osteoblasts and osteoclasts [31]. In addition, bone cells share the bone marrow microenvironment with cells from hematopoietic and mesenchymal lineage, which interact mainly through secreted factors to maintain bone homeostasis. The bone impairment observed in patients with cystinosis may therefore be the result of a combined effect of intrinsic cystinosin deficiency on one hand, and impaired secretion of communication molecules exerting pro- or trans- differentiation cues on neighboring cells on the other hand. Even though the second part of the later sentence remains hypothetical, there is current evidence to demonstrate intrinsic biological effects of cystinosin deficiency both in osteoblasts and in osteoclasts.

Conforti et al. showed in 2015 that mesenchymal stem cells (MSCs) isolated from a patient with cystinosis display a reduced ability to differentiate into osteoblasts, which can be reverted after cysteamine treatment [22]. Therefore, cystine efflux mediated by cystinosin seems to be important for MSCs derived osteoblast differentiation. Later on, we showed that in vitro low doses of cysteamine (i.e., 50 µM) also stimulate murine osteoblastic differentiation and mineralization, with an inhibitory effect at higher doses (i.e., 200 µM), likely explaining, at least partly, the bone toxicity observed in patients receiving cysteamine dose over 50µM. The BrdU labeling showed a significantly lower cell proliferation in the 200 µM group, but a cytotoxic effect of cysteamine was ruled out since no increased LDH levels were found in the supernatants at the different cysteamine concentrations [23].

Battafarano et al. also recently described osteoblastic dysfunction in Ctns^-/-^ mice [30]: the expression of cystinosin was first evaluated in cultures of wild-type MSCs, and increased during osteoblast differentiation. Further analyses showed that osteoblastic cultures from Ctns^-/-^ mice displayed increased cystine content and a decreased number of alkaline phosphatase positive cells, with a decreased expression of the main genes involved in osteoblastic differentiation and activity (and notably Runx2, ALP, and Col1A2). In vitro mineralization assays also demonstrated a reduced ability to release mineralized nodules from cystinotic osteoblasts as compared to control cells, although this study did not address the in vivo therapeutic effect of cysteamine on bone phenotype [30].

Second, in human osteoclasts differentiated from peripheral blood mononuclear cells (PBMCs), we showed that CTNS is required for proper osteoclastic differentiation and resorption activity, with a progressive increase of *CTNS* gene expression during osteoclastogenesis, with a peak observed at Day 6 on mature osteoclasts following the same pattern that cathepsin K transcripts a late marker of osteoclast differentiation which is essential for bone resorption [23]. These expressions are under the control of monocyte colony-stimulating factor (M-CSF) and receptor activator of nuclear factor kappa-B ligand (RANKL), the two major cytokines driving osteoclast differentiation and function. Indeed, when osteoclastogenesis was examined in basic conditions without any treatment, an increase of low nuclei number TRAP (tartrate-resistant acid phosphatase) positive cells was observed in patients in comparison to controls, despite the fact that they were generated from the same number of monocyte progenitors. These findings indicate that cystinosin deficiency favors osteoclastogenesis and that cystinosin-lacking mononuclear progenitors are more prone to generate osteoclasts than controls. 

We therefore evaluated resorption capacity of osteoclasts derived from cystinotic patients; in contrast to what we observed for differentiation, in vitro bone resorption is decreased in osteoclasts from patients as compared to controls; similarly, we also found that cathepsin K transcript levels are decreased in patients in comparison to controls. Overall, in vitro, cystinosin deficiency increases osteoclastogenesis but impairs bone resorption activity of osteoclasts derived from human PBMCs.

The analysis of cysteamine treatment on both osteoclastic differentiation and resorption: cysteamine reveals no effect on osteoclastic differentiation whatever the origin of the donors and the doses tested. However, when examining the resorptive activity of cysteamine-treated osteoclasts both in controls and patients, unexpectedly, a beneficial anti-resorptive effect is observed in controls, showing a 50% inhibition of the resorption activity with 50 µM. In contrast, resorption by patient-derived osteoclasts is not changed at this concentration, the decreased osteoclastic activity is detectable and significant only at 200 µM of cysteamine [23]. To summarize, in vitro, neither the defect in differentiation nor the defect in resorption of CTNS patient-derived osteoclasts is rescued by cysteamine treatment, at any concentration. Even though cystine cell content was not measured, these results suggest that neither osteoclast differentiation nor activity are dependent on cystine efflux from lysosomes but may instead rely on cystinosin itself or its interaction with other cellular pathways.

It is noteworthy that Battafarano et al. observed almost similar results in murine cells: a progressive increase of the *Ctns* gene expression was observed during osteoclastogenesis. In contrast, after differentiation, a lower number of mature osteoclasts was observed in Ctns^-/-^ cultures compared with wild-type cells. This discrepancy with regards to human osteoclast cultures could be due to the fact that murine osteoclasts, in this case, are derived from the bone marrow of mice with no renal failure. Furthermore, a reduced ability to resorb bone was also seen in *Ctns*^-/-^ osteoclasts [30].

Lysosomes, vesicular trafficking and exocytosis are essential and tightly regulated in mature osteoclasts in order to achieve an efficient resorption [32]. Among human autosomal lysosomal storage diseases, some were reported to display a bone phenotype secondary to mutation of lysosomal- and osteoclast-shared genes, such as Vacuolar H+ ATPases, *TCIRG1*, *ClCN7*, *OSTM1*, *SLC29A3*, Cathepsin K and *MMP9*, all of them leading to an osteopetrotic or osteosclerotic bone phenotype due to impaired osteoclast mediated resorption, and bone hardening with increased bone mineral density [33]. Thus, depending on the underlying mutation in patients with cystinosis, different degrees of osteoclastic dysfunction may be hypothesized. This working hypothesis nevertheless deserves future studies.

In conclusion, cystinosin deficiency induces an intrinsic defect both in osteoblasts and osteoclasts; moreover, cysteamine dosage exerts dual effects on bone cells: in vitro low concentration of cysteamine have beneficial anti-resorptive effects on healthy human derived osteoclasts, but cysteamine treatment does not rescue *Ctns^-/-^* defects neither at the differentiation nor at the resorption level. Our study reveals that cystinosin holds a function in osteoclasts that is independent of lysosomal cystine efflux. 

On the contrary, low doses of cysteamine stimulate osteoblastic differentiation and mineralization, with an inhibitory effect at higher doses, likely explaining, at least partly, the bone toxicity observed in patients receiving high doses of cysteamine [23], and the ability of cysteamine treatment to rescue osteoblastic differentiation of human *CTNS^-/-^* MSCs [21]. Figure 1 summarizes the current knowledge of the effects of cystinosin deficiency and cysteamine therapy in osteoblasts and osteoclasts. The next step that is clearly more physiological would integrate in a same experiment the two cellular components of bone homeostasis to evaluate the crosstalk between osteoblasts and osteoclasts.

### 1.4. Future Research Axes: Toward Damaged Mitochondrial Function in Osteoclasts Derived From Patients with Nephropathic Cystinosis?

Recent findings demonstrated that lysosomal dysfunction in cystinosis is associated with defective autophagic mitochondria clearance in epithelial cells from the proximal tubule. These defects lead to increased oxidative stress that disorganizes tight-junction and might explain the epithelial defects in cystinosis [34]. In CTNS deficient cells, several studies report increased mitochondrial fragmentation and a deregulation of the turnover, of parkin, mitofusin2 and fission1 proteins that are proteins involved in mitochondrial dynamics. Correction of these defects provide new therapeutic perspectives as cysteamine only rescue partly these defects [35,36]. In line with these observations, osteoclasts are multinucleated giant cells that contain a very high density of mitochondria, and increased mitochondria biogenesis and oxidative phosphorylation increase bone resorption. Interestingly enough, very recent studies revealed that mitofusin 2 facilitates osteoclastogenesis by modulating calcium -calcineurin -NFATc1 (Nuclear factor of activated T-cell cytoplasm1); thus, it is tempting to propose that mitochondria function might be damaged in CTNS deficient osteoclasts [37,38]. 

### 1.5. Future Research Axes: Toward an Impaired mTor Signaling in Osteoclasts Derived from Patients with Nephropathic Cystinosis?

The study of proximal tubular kidney cells obtained from healthy donors and patients with cystinosis reveals that cystinosin deficiency is associated with altered mammalian target of rapamycin complex 1 (mTORC1) signaling, characterized by abnormal lysosomal retention of mTORC1 during starvation followed by delayed reactivation of mTORC1 [39]. Alteration of the mTORC1 signaling pathway can therefore participate in the development of the proximal tubular dysfunction that characterizes nephropathic cystinosis. Interestingly, this alteration is not reversed by the administration of cysteamine [39]. In parallel, mTORC1 signaling also plays an important role in skeletal development and bone remodeling: mTORC1 activity is down-regulated during osteoclastic differentiation through the negative regulator TSC1 (tuberous sclerosis complex 1), whose absence impairs RANKL-dependent osteoclastogenesis [40]. It is possible that the mTORC1 pathway is involved in bone impairment in cystinosis. Our findings indicate that defects of *Ctns^-/-^* osteoclasts might not rely on cystine efflux but on cystinosin itself or its interaction with other pathways, as recently discussed in a general review of new perspectives in cystinosis [41]. In this view, one can speculate that cystinosin interacts with mTORC1 at the lysosome membrane to modulate osteoclastogenesis.

Andrzejewska et al. showed that the mTORC1 pathway is down-regulated in proximal tubular cell lines derived from *Ctns^-/-^* mice [42]. A similar down-regulation in human osteoclastic progenitors could account for the overall increased osteoclastogenesis that we observe in cystinosis patients, as compared to controls. Andrzejewska et al. also demonstrated that cystinosin interacts with the vacuolar H+-ATPase-Ragulator-Rag complex which controls mTORC1 localization to lysosomes and thus mTORC1 signaling [42]. 

DNA mutations in the *CTNS* gene have various functional consequences, linked to their structural impact on cystinosin. Extensive deletions (such as the 57-kb deletion) cause the absence of protein while severe truncating mutations lead to the synthesis of an inactive variant [11]. Both these situations amount to a loss of cystinosin efflux function. In contrast, milder mutations allow the synthesized cystinosin variant to retain residual activity. It is interesting to hypothesize that as well as canceling cystinosin efflux function, severe *CTNS* mutations impair the interaction between the Ragulator-Rag complex and mTORC1, preventing its activation. On the other hand, mutations of more limited structural impact might allow, in some extent, to maintain an efflux activity as well as the interaction between mTORC1 and the lysosomal membrane-attached Ragulator-Rag complex.

It could be argued that the down-regulation of mTORC1 is due to the accumulation of cystine. However, as shown by Andrzejewska et al., decrease of lysosomal cystine levels by cysteamine did not rescue mTORC1 activation in proximal tubular cells, thus suggesting that the down-regulation of mTORC1 is due to the absence of cystinosin rather than to the accumulation of cystine [42]. 

Even though this remains a working hypothesis that deserves to be confirmed, Figure 2 illustrates the hypothetical mechanisms of mTORC1 deregulation in osteoclasts derived from patients with cystinosis.

## 2. Conclusions

Bone impairment in cystinosis is now recognized as a late-onset complication of a multi-systemic disease. It is important that physicians take these potential complications into account in the management of patients with nephropathic cystinosis, as they can considerably affect their quality of life and even require invasive procedures. International guidelines were recently published to improve the diagnosis and management of CMBD, although the underlying pathophysiology remains to be fully elucidated. In the future, understanding the differential impact of the different *CTNS* mutations and/or cysteamine therapy on bone cell biology and the molecular mechanisms behind these differential effects should lead to more targeted therapeutic strategies.

## Figures and Tables

**Figure 1 ijms-21-03109-f001:**
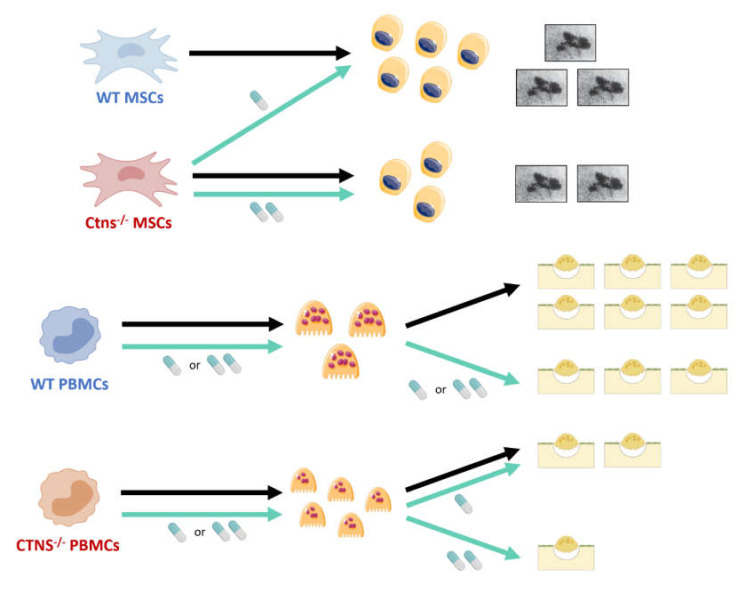
Current understanding of osteoblastic and osteoclastic defects in cystinosis, resulting from a combination of cystinosin deficiency and cysteamine effects. Mesenchymal stem cells (MSCs) isolated from a patient with cystinosis display a reduced ability to differentiate into osteoblasts, which can be reverted after cysteamine treatment. In vitro mineralization assays demonstrate a reduced ability to release mineralized nodules from cystinotic osteoblasts as compared to control cells. In vitro low doses of cysteamine (50 µM) stimulate osteoblastic differentiation and mineralization, with an inhibitory effect at higher doses (200 µM). Cystinosis favors osteoclastogenesis, mononuclear progenitors being more prone to generate osteoclasts than controls. Cysteamine has no effect on osteoclastic differentiation whatever the origin of the donors and the doses tested. However, it is noteworthy that cysteamine displays a beneficial anti-resorptive effect: in controls, a 50%-inhibition of resorption activity is observed with 50µM. As for patients, the decreased osteoclastic activity is detectable and significant only at 200 µM. WT, Wild-type; Ctns^-/-^, gene knockout (murine model); CTNS^-/-^, cystinotic patients; MSC, Mesenchymal stem cell; PBMC, Peripheral blood mononuclear cell. Black arrow: no treatment; Blue arrow: cysteamine treatment. Pill icon: cysteamine - 50 µM (one pill), 200µM (two pills). Top right pictures: mineralization nodules magnification (Von Kossa staining); Bottom right illustrations: osteoclastic resorption pits.

**Figure 2 ijms-21-03109-f002:**
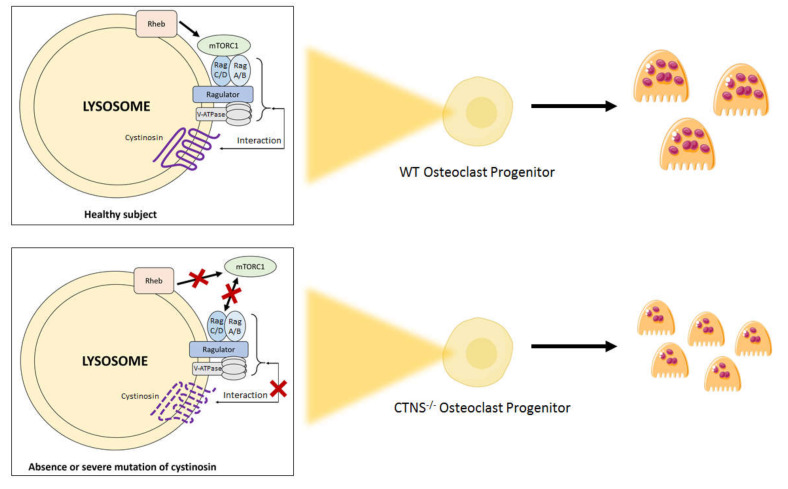
Hypothetical regulation of mtorc1 activation in osteoclasts derived from healthy donors and from patients with nephropathic cystinosis. The Ragulator-Rag Complex has a key role in mTOR signaling regulation. The Ragulator, along with the Rag GTPases, are necessary for the localization of mTORC1 to the lysosome surface. When enough amino acids are present, Rag GTPases become activated, which leads to the translocation of mTORC1 from the cytoplasm to the lysosome surface. This process allows mTORC1 to bind to Rheb, which induces the kinase activity of mTORC1. Absence or severe mutation of cystinosin, a protein interacting with the Ragulator-Rag complex, might prevent the activation of mTORC1 by Rheb, which would increase the final number of osteoclasts at the end of the differentiation process. mTORC1, mammalian target of rapamycine complex 1.

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
