# Peer review of "Bone Disease in Nephropathic Cystinosis: Beyond Renal Osteodystrophy"

_ijms, 2020, doi:10.3390/ijms21093109_

Round 1
Reviewer 1 Report
The authors show in this manuscript a clinical update on bone disease in cystinosis with the latest published data, in part described in the recent report for management of CMBD (Ref. 21).
Then authors, although the available literature is limited, clearly explain the unbalanced osteoblastic and osteoclastic processes in cystinosis and the effects of cysteamine.
It is interesting and convincing the theorized role of mTORC1 pathway in osteoclastogenesis, however authors could enhance the contents of the manuscript by analyze also the emerging role of mitochondria in cystinosis related to their significant function in bone homeostasis.
Minor comments:
- On paragraph 1.1 “cysteamine enters the lysosome and breaks cystin disulfide bond, resulting in the formation of cysteine and a cysteamine disulfide. These can then leave the lysosome via the cysteine transporter and PQLC2, respectively. Cysteamine therefore restores the efflux of cystin by reacting with it, products of the reaction being able to bypass the deficient transporter (13,14)”. cystin should be replaced with cystine; cysteamine disulfide should be replaced with cysteamine-cysteine mixed disulfide.
- Labels within the figure 2 should be enlarged to be more readable
Author Response
The authors show in this manuscript a clinical update on bone disease in cystinosis with the latest published data, in part described in the recent report for management of CMBD (Ref. 21). Then authors, although the available literature is limited, clearly explain the unbalanced osteoblastic and osteoclastic processes in cystinosis and the effects of cysteamine. It is interesting and convincing the theorized role of mTORC1 pathway in osteoclastogenesis, however authors could enhance the contents of the manuscript by analyze also the emerging role of mitochondria in cystinosis related to their significant function in bone homeostasis.
Thank you for this overall positive appreciation of our manuscript. We have briefly discussed the emerging role of mitochondria in cystinosis, as follows:“Recent findings have demonstrated that lysosomal dysfunction in cystinosis is associated with defective autophagic mitochondria clearance in epithelial cells from the proximal tubule. These defects lead to increased oxidative stress that disorganizes tight-junction and might explain the epithelial defects in cystinosis. In CTNS deficient cells, several studies report increased mitochondrial fragmentation and a deregulation of the turn-over, of parkin, mitofusin2 and fission1 proteins, that are proteins involved in mitochondrial dynamics. Correction of these defects provides new therapeutic perspectives as cysteamine only rescue partly these defects. In line with these observations, osteoclasts are multinucleated giant cells that contain a very high density of mitochondria, and increase mitochondria biogenesis and oxidative phosphorylation increase bone resorption. Interestingly enough, very recent studies revealed that mitofusin 2 facilitates osteoclastogenesis by modulating calcium -calcineurin -NFATc1 (Nuclear factor of activated T-cell cytoplasm1); thus it is tempting to propose that mitochondria function might be damaged in CTNS deficient osteoclasts.”.
Minor comments:
- On paragraph 1.1 “cysteamine enters the lysosome and breaks cystin disulfide bond, resulting in the formation of cysteine and a cysteamine disulfide. These can then leave the lysosome via the cysteine transporter and PQLC2, respectively. Cysteamine therefore restores the efflux of cystin by reacting with it, products of the reaction being able to bypass the deficient transporter (13,14)”. cystin should be replaced with cystine; cysteamine disulfide should be replaced with cysteamine-cysteine mixed disulfide.
Thank you for these comments. We have modified accordingly.
- Labels within the figure 2 should be enlarged to be more readable
Thank you for pointing out this problem. The figure has been reworked.
Reviewer 2 Report
P2 If on cysteamine, few pateints should have reached ESR by < 14 years
P2 After 3.9 years the incidence of fx had decreased to 12%. Why? Is the risk in the general population 4%? Tie this to cystinosis.
P4 Which strain of Ctns-/- mice were used, the earlier that did not develop renal failure or the inbred which did?
P5 "...although for patients the decreased osteoclastic activity is detectable and significant only at 200 µM of cysteamine (23)." Patient blood levels rarely rise to 100µM, and never to 200µM, therefore the implication to occurences in patients is erroneous
P5What were the cystine levels in these cells, and how long were they exposed to cysteamine?How was the cystine quantified? mAbsewnt these data one cannot conclude “ neither osteoclast differentiation nor activity are dependent on cystine efflux from lysosomes…”
P7 "...Our findings show that defects of Ctns-/- osteoclasts do not rely on cystine efflux but on cystinosin itself or its interaction with other pathways." This requires data showing cystine depletion in the osteoclasts.
P7 Ref 37 is mis-numbered as 38.
P8 "Bone impairment in cystinosis is a late-onset complication of a multi-systemic disease." Should speculate on why it is a late-appearing complication.
Author Response
Page 2 : If on cysteamine, few patients should have reached ESRD by < 14 years.
Thank you for this comment. The main aim of the introduction of this review paper was to “set the floor” and to remind the reader of the main concepts of bone osteodystrophy in pediatric chronic kidney disease, before addressing the specific concept of “cystinosis metabolic bone disease, CMBD”. A such, even though we agree that the onset of ESRD in cystinosis (at least in developed countries in compliant patients….) is rather observed during the second and third decade of life (i.e., teenage and young adultohood), we do not feel that this is inadequate to introduce the concept of ROD in children reaching ESRD before 14 years of age…
Page 2: After 3.9 years, the incidence of fractures had decreased to 12%. Why? Is the risk in the general population 4%? Tie this to cystinosis.
Thank you this comment. Again, the introduction is here to “set the floor” and we will not discuss these data that have been published by well-known teams in high impact journals… The main objective there was to show that all pediatric patients with CKD display a significant risk of bone fracture, that is estimated to be 2 to 3 times higher than in the general population. These numbers are important, since we discuss later specific data in cystinosis: as such readers not very well aware of bone disease in pediatric CKD and in cystinosis will be able to understand this higher bone risk in cystinosis.
The link with cystinosis already appeared in the previous version of the manuscript in the section entitled “Bone impairment in patients nephropathic cystinosis: clinical facts”. More generally, the last paragraph of the introduction ties all the data discussed above to orphan renal diseases inducing bone impairment, and more specifically to cystinosis, as follows: “Moreover, in addition to “conventional” ROD and CKD-MBD, there is now evidence that some genetic renal diseases may also have a direct negative impact on bone per se, notably in primary hyperoxaluria, nephropathic cystinosis, autosomal dominant polycystic kidney disease, and Pierson syndrome. The main objective of this manuscript is not only to provide a clinical update on bone disease in cystinosis, but also to summarize the current experimental evidence clearly demonstrating a functional impairment of bone cells in this orphan disease and to discuss new working hypotheses that deserve future research in the field.”.
Page 4: Which strain of Ctns-/- mice was used, the earlier that did not develop renal failure or the inbred which did?
Thank you for raising this very important point. The first global Ctns knock-out mice model was generated by S. Cherqui et al , 2002 (ref 25) in C57BL/6 mice back-ground: we have added this detail in the text ( line 180). These authors were the first to report no renal failure in young animals (until 6month-old mice) but bone cortical defects that were visible by X-ray and histology in 8 month old mice.
The same mice model was used by Battafarano et al 2019 (ref 31). They found in 1 month old Ctns-/- C57BL/6 mice, skeletal alterations, ie reduction of trabecular bone volume, decreased BMD , cortical thickness and cellular parameters. Kidney phenotype was assessed in male animals at different ages: 1, 3 and 6 months. In this study too, incipient renal disease was seen only at 6 months, indicating that the observed bone phenotype in 1 month old mice was mainly due to intrinsic defect of Ctns-/- bone cells. In contrast, they reported that the bone phenotype disappears in 3 and 6 months old mice. However, cortical defects were detected in 6 month old mice, concurrent to renal disease onset.
In the previous version of the manuscript (line 183), we already mentioned that “The study of bone phenotype in one month-old Ctns knockout (Ctns-/-) mice reveals a decrease in bone mineral density (BMD), trabecular volume (TV) and number/thickness of bone trabeculae compared to control animals. Incidentally, these mice did not display any sign of Fanconi syndrome.”. We have now added the C57BL/6 background.
To our knowledge, the bone phenotype was not analyzed in the inbred or any other existing Ctns deficient model.
Page 5: Patient blood levels rarely rise to 100µM and never to 200µM, therefore the implication to occurrences in patients is erroneous.
Thank you for this comment. We fully agree with the reviewer, but we never stated that results obtained in vitro with 200µM of cysteamine were directly transposable as such in patients. Our sentence (line 208-210) was extremely cautious and moderate “likely explaining, at least partly.”
We have though modified it accordingly.
Page 5: What were the cystine levels in these cells, and how long were they exposed to cysteamine? How was the cystine quantified? Absent these data, one cannot conclude “neither osteoclast differentiation nor activity are dependent on cystine efflux from lysosomes…”.
We respectfully remind the reviewer that the manuscript is not an original report but a review, in consequence, the experimental details of all original studies quoted in this manuscript are obviously not reported. Indeed, cysteamine treatment is described in the materials and methods section in the original paper (ref 23), ie the cysteamine treatment was added from the second day on of PBMC plating and maintained all along the differentiation. Cystine was not assessed in the quoted study and this point was added to the text (line 247).
Page 7: “…Our findings show that defects of Ctns-/- osteoclasts do not rely on cystine efflux but on cystinosin itself or its interaction with other pathways.” This requires data showing cystine depletion in the osteoclasts.
This comment is related to the previous one. We respectfully disagree with this remark; as p7 is part of the section 1.4 axes for future research axes , where typically, in a review process, one is supposed to propose and open new perspective for cystinosin function that would be different from conventional ‘ cystine efflux pathway”. These new putative pathways are also illustrated in a recent short commentary from Pr Langman, a world-known expert on the disease, and entitled “Oh cystinosin: let me count the ways !” in Kidney International (2019). We have added this reference to the manuscript.
Cystine depletion was not assessed, but here are the facts: cysteamine treatment was applied and had no effect at any concentration on patient cells to rescue them, either to a better differentiation or higher resorption activity. This indicates that at least the osteoclast deficiency does not rely on cystine efflux or other cellular side effect of cysteamine , but on the cystinosin deficiency it-self.
Therefore we have kept the idea in the text, even though we have nuanced line 363, as follows: “Our findings indicate ( instead of show) that defects of Ctns-/- osteoclasts might (instead of do) not rely on cystine efflux but on cystinosin itself or its interaction with other pathways”.
Page 7: Ref. 37 is mis-numbered as 38.
Thank you for pointing out this problem. We have corrected accordingly.
Page 8: “Bone impairment in cystinosis is a late-onset complication of a multi-systemic disease.” Should speculate on why it is a late-appearing complication.
Thank you for this remark. This comment corresponds to the summary statement we provide in the conclusion of the paper to put in perspective the experimental data we have presented to the clinical phenotype. This statement of “late-onset complication” is in line with the recent international guidelines published on diagnosis and management of bone that define CMBD as follows “The metabolic consequences include hypophosphatemic rickets and growth failure. Later in life, individuals with cystinosis also suffer from mineral and bone disorder related to chronic kidney disease (CKD-MBD), including renal osteodystrophy, resulting in a complex bone phenotype termed Cystinosis Metabolic Bone Disease (CMBD)”. Again, in this review paper (and not an original paper) we dot not feel adequate to justify this concept that has emerged from an international consensus of experts of the disease.
Moreover, we already discussed the current hypotheses to explain such a pathophysiology (lines 95-103), as follows: “Even though its exact underlying pathophysiology remains unclear, at least five distinct but complementary entities can explain CMBD in addition CKD-MBD and post-transplant CKD-MBD: 1/ long-term consequences of hypophosphatemic rickets and renal Fanconi syndrome (i.e., hypophosphatemia, metabolic acidosis, chronic hypovolemia, hypocalcemia and 1-25 OH2 vitamin D deficiency) together with iatrogenic effects of its supportive management, 2/ deficiency in nutrition and micro-nutrition, and notably copper deficiency, 3/ hormonal disturbances such as hypothyroidism, hypogonadism, hypoparathyroidism and resistance to growth hormone and IGF1, 4/ myopathy, and 5/ intrinsic and iatrogenic bone lesions such as direct bone effects of CTNS mutation and cysteamine on osteoblasts and osteoclasts”. From this list of putative factors, some of them are clearly considered as “late-complications” of cystinosis: the reviewer may be well aware that endocrinological complications and hypogonadism appear at the end of teenage and adulthood…
We have modified as follows:” Even though its exact underlying pathophysiology remains unclear, at least five distinct but complementary entities can explain CMBD in addition CKD-MBD and post-transplant CKD-MBD: 1/ long-term consequences of hypophosphatemic rickets and renal Fanconi syndrome (i.e., hypophosphatemia, metabolic acidosis, chronic hypovolemia, hypocalcemia and 1-25 OH2 vitamin D deficiency) together with iatrogenic effects of its supportive management, 2/ deficiency in nutrition and micro-nutrition, and notably copper deficiency, 3/ hormonal disturbances such as hypothyroidism, hypogonadism and hypoparathyroidism usually appearing in late teenage or early adulthood, and resistance to growth hormone and IGF1, 4/ myopathy, and 5/ intrinsic and iatrogenic bone lesions such as direct bone effects of CTNS mutation and cysteamine on osteoblasts and osteoclasts
Reviewer 3 Report
This review gives a very good update on bone pathology in nephropathic cystinosis, a very rare inherited metabolic lysosomal storage disorder resulting in kidney disfunction and among other abnormalities, bone deformities and fragility. The authors provide an overview on the recent literature of specific bone cell abnormalities in the disorder.
The review is well-written and I have only minor comments:
There are many abbreviations throughout the text, and it would be very helpful to add an abbreviation list.
1.1 Please add the prevalence of the disorder
There is a review from Cherqui S and Courtoy PJ, 2017 that should be cited.
1.2 The last paragraph on page 3 is very dense. The authors should consider some rewording in order to make the text more accessible to a non-specialized readership.
1.3 Third paragraph of page 5: “Second, in human osteoclasts ……” there is a reference missing (probably Ref: 23: Clabarmunt-Tabernet et al., Nephrol Dial Transplant 2018. This reference needs also to be updated in the Ref. list)
1.4 Figure 2. The font and the cartoons of the lysosomes are very small. Please provide a legible figure.
Author Response
This review gives a very good update on bone pathology in nephropathic cystinosis, a very rare inherited metabolic lysosomal storage disorder resulting in kidney disfunction and among other abnormalities, bone deformities and fragility. The authors provide an overview on the recent literature of specific bone cell abnormalities in the disorder. The review is well-written and I have only minor comments.
We would like to thank you for this overall positive appreciation of the manuscript.
There are many abbreviations throughout the text, and it would be very helpful to add an abbreviation list.
Thank you for this comment. We propose a Table 1, summarizing all these abbreviations, as follows.
|
Bone impairment in chronic kidney disease CKD : chronic kidney disease CKD-MBD : chronic kidney disease mineral and bone disorders GH : growth hormone IGF1 : insulin-like growth factor 1 ROD : renal osteodystrophy PTH : parathyroid hormone |
|
Bone involvement in cystinosis CTNS : lysosomal cystine transporter CMBD : cystinosis metabolic bone disease |
|
Clinical observations of bone impairment in nephropathic cystinosis ALP : alkaline phosphatase 25-D : 25 OH-vitamin D FGF23 : fibroblast growth factor 23 DXA : dual X-ray absorptiometry HR-pQCT : high resolution peripheral quantitative computed tomography BMD : bone mineral density |
|
Impairment of bone cells in nephropathic cystinosis TV : trabecular volume MSC : mesenchymal stem cell BrdU : bromodeoxyuridine LDH : lactate dehydrogenase RUNX2 : runt-related transcription factor 2 COL1A2 : collagen type I alpha 2 chain M-CSF : macrophage colony-stimulating factor RANKL : receptor activator of nuclear factor kappa-B ligand TRAP : tartrate-resistant acid phosphatase PBMC : peripheral blood mononuclear cell TCIRG1 : T-cell immune regulator 1 ClCN7 : chloride voltage-gated channel 7 OSTM1 : osteopetrosis associated transmembrane protein 1 SLC29A3 : solute carrier family 29, member 3 MMP9 : matrix metallopeptidase 9 |
|
Hypothetical mechanism of mTORC1 dysregulation in osteoclasts mTORC1 : mammalian target of rapamycin complex 1 TSC1 : tuberous sclerosis complex 1 |
1.1 Please add the prevalence of the disorder.There is a review from Cherqui S and Courtoy PJ, 2017 that should be cited.
Thank you for this comment. We have added as follows: “Cystinosis, affecting 1 to 9 / 100 000 persons, accounts for 1.4% of children on dialysis and 2.1% of pediatric renal transplant recipients in the USA (Cherqui Courtoy 2017)”
1.2 The last paragraph on page 3 is very dense. The authors should consider some rewording in order to make the text more accessible to a non-specialized readership.
Thank you for this comment. We have tried to shorten the paragraph, as follows: “Using HR-pQCT, significant decreases in cortical parameters and total bone mineral density were observed in cystinotic patients in comparison to controls at the tibia. There were no differences for trabecular parameters. Interestingly, there were no obvious abnormalities of circulating and urinary calcium as well as ALP levels, but biochemical tests showed a tendency toward low PTH and low FGF23 levels, likely reflecting chronic phosphate wasting because of low phosphate reabsorption rates, although there was no obvious uncontrolled Fanconi syndrome in patients with conservative management and in those after transplantation. When comparing patients with a functioning renal graft, spine Z-scores were significantly greater in patients with a past of nephrectomy in comparison to patients without nephrectomy of native kidneys. We concluded that chronic phosphate wasting may silently impair bone quality in cystinosis, since nephropathic cystinosis is a unique state of long-term CKD-MBD and renal phosphate wasting due to the Fanconi syndrome that persists even at late CKD stages. These preliminary data still need to be confirmed in larger cohorts, since this would provide a strong rationale for proposing nephrectomy of native kidneys to patients in order to decrease renal phosphate wasting and therefore improve bone quality.”. We hope that it is now more easily readable.
1.3 Third paragraph of page 5: “Second, in human osteoclasts ……” there is a reference missing (probably Ref: 23: Clabarmunt-Tabernet et al., Nephrol Dial Transplant 2018. This reference needs also to be updated in the Ref. list)
We have corrected accordingly and apologize for the mistake.
1.4 Figure 2. The font and the cartoons of the lysosomes are very small. Please provide a legible figure.
Thank you for pointing out this problem. The figure has been reworked.
Reviewer 4 Report
The group has provided a useful service for readers in unpicking the diversity of factors contributing to a rare renal condition and its treatment and proposing the interrelationship between them.
The review is comprehensive, combining a depth of personal interest and previous publication on the topic, with the breadth provided by others cited in the field. Their hypothesis is justified by the evidence from patients and animal models - their own contribution placed in the context of others.
1.The strength lies in the two figures (though in Figure 1 the magnification of the mineralized nodules is essential in the legend, and some of the text included in Figure 2 is too small to read). Nevertheless, these figures show that a good diagram is better than a thousand words. Clearly much thought has gone into them and they are highly commended.
2. I am a biologist and not a clinical specialist. For me, the text seems excessively dense with facts and a complexity of comparisons together with a distracting overload of unfamiliar abbreviations that required reading and checking and re-reading and losing track of the main theme in the process. The detail is relevant and the general structure is logical; therefore to make the text more concise and easily accessible a table is essential summarising each prime aspect of background and hypothesis together with ALL abbreviations clearly defined at a glance in one place without having to hunt through the script repeatedly for clarification.
3. It is some time since I was involved in the histological investigation of renal osteodystrophy. I recall extensive osteomalacia due to deficient active vitamin D metabolites and well-defined resorption cavities occupied by active osteoclasts due to secondary hyperparathyroidism. I was therefore unsettled (again by all the abbreviations perhaps) to see hypoparathyroidism listed as a "complementary entity" to cystinosis metabolic bone disease (p3, 1st para).
4. In describing the pilot investigation performed by the authors (p4, 1st para) it is briefly mentioned that the bone impairment of cystinosis in the youthful subjects (and a mouse model) impacts upon cortical rather than trabecular bone which of course is usually more vulnerable with its larger surface are for bone cell activity and functional impairment. This seems such a key observation of fundamental consequence (e.g. concerning intramembranous versus endochondral bone behaviour) that to my mind it merits more prominance (maybe even a mention in the Abstract?).
5. Editorially there is some repetition. For example, the final sentence of the Abstract is repeated verbatim as the final sentence of the Introduction. Also the word "clearly" is overused throughout without justification, as is the meaningless (in this context) word "indeed". Again "dramatically" appears at least twice and has no place at all in good science, and there is no such word as "combinatory" (p4, last para), nor "dysregulation" (p7, bottom line), and "crosstalk" is not in my dictionary. There is also a tendency to overstate by adding adjectives e.g. "very," "considerably" and "paramount importance" (p.8) when paradoxically understatement can be more compelling.
6. The concluding paragraph is carelessly written in comparison with what has gone before and rather than stamping authority on the review it diminishes it, leaving the reader short-changed and dubious. For example, the device "on the one hand" is a useful one but must be partnered with "on the other hand" or it is nonsensical.
Author Response
The group has provided a useful service for readers in unpicking the diversity of factors contributing to a rare renal condition and its treatment and proposing the interrelationship between them. The review is comprehensive, combining a depth of personal interest and previous publication on the topic, with the breadth provided by others cited in the field. Their hypothesis is justified by the evidence from patients and animal models - their own contribution placed in the context of others.
We would like to thank you for this overall positive appreciation of the manuscript.
1.The strength lies in the two figures (though in Figure 1 the magnification of the mineralized nodules is essential in the legend, and some of the text included in Figure 2 is too small to read). Nevertheless, these figures show that a good diagram is better than a thousand words. Clearly much thought has gone into them and they are highly commended.
We would like to thank you warmly for this specific comment.
- I am a biologist and not a clinical specialist. For me, the text seems excessively dense with facts and a complexity of comparisons together with a distracting overload of unfamiliar abbreviations that required reading and checking and re-reading and losing track of the main theme in the process. The detail is relevant and the general structure is logical; therefore to make the text more concise and easily accessible a table is essential summarising each prime aspect of background and hypothesis together with ALL abbreviations clearly defined at a glance in one place without having to hunt through the script repeatedly for clarification.
Thank you for this comment. We propose a Table 1, summarizing all these abbreviations, as follows.
|
Bone impairment in chronic kidney disease CKD : chronic kidney disease CKD-MBD : chronic kidney disease mineral and bone disorders GH : growth hormone IGF1 : insulin-like growth factor 1 ROD : renal osteodystrophy PTH : parathyroid hormone |
|
Bone involvement in cystinosis CTNS : lysosomal cystine transporter CMBD : cystinosis metabolic bone disease |
|
Clinical observations of bone impairment in nephropathic cystinosis ALP : alkaline phosphatase 25-D : 25 OH-vitamin D FGF23 : fibroblast growth factor 23 DXA : dual X-ray absorptiometry HR-pQCT : high resolution peripheral quantitative computed tomography BMD : bone mineral density |
|
Impairment of bone cells in nephropathic cystinosis TV : trabecular volume MSC : mesenchymal stem cell BrdU : bromodeoxyuridine LDH : lactate dehydrogenase RUNX2 : runt-related transcription factor 2 COL1A2 : collagen type I alpha 2 chain M-CSF : macrophage colony-stimulating factor RANKL : receptor activator of nuclear factor kappa-B ligand TRAP : tartrate-resistant acid phosphatase PBMC : peripheral blood mononuclear cell TCIRG1 : T-cell immune regulator 1 ClCN7 : chloride voltage-gated channel 7 OSTM1 : osteopetrosis associated transmembrane protein 1 SLC29A3 : solute carrier family 29, member 3 MMP9 : matrix metallopeptidase 9 |
|
Hypothetical mechanism of mTORC1 dysregulation in osteoclasts mTORC1 : mammalian target of rapamycin complex 1 TSC1 : tuberous sclerosis complex 1 |
- It is some time since I was involved in the histological investigation of renal osteodystrophy. I recall extensive osteomalacia due to deficient active vitamin D metabolites and well-defined resorption cavities occupied by active osteoclasts due to secondary hyperparathyroidism. I was therefore unsettled (again by all the abbreviations perhaps) to see hypoparathyroidism listed as a "complementary entity" to cystinosis metabolic bone disease (p3, 1st para).
Thanks for this comment. In the specific setting of cystinosis, hypopara (or relative hypopara), as well hypo-FGF23 (or relative hypo-FGF23) seems to be a response to renal phosphate wasting that can be seen even at late CKD stages. In this view, it would be rather hypophosphatemia that would be deleterious for bone, as well as the other components of proximal tubulopathy (such as acidosis).
Concerning your specific point of adynamic bone disease (rather than extensive osteomalacia) in end-stage renal disease induced by high doses of active vitamin D analogs and high doses of calcium (you may refer to the papers published by Drs Salusky and Goodman in pediatric CKD), we may say that you are right, but it was seen a few years ago, and we do not see so many patients with adynamic bone disease in pediatrics anymore. Indeed, the problem is rather on secondary hyperparathyroidism and the risk of osteitis fibrosa with indeed active osteoclast, resorption and disorganized mineralization. However, with the novel therapies, SHPT is probably better controlled in 2020 than it was a few years ago (nutritional phosphate control, new phosphate binders, vitamin D analogs but also calcimimetics, etc). It is out of the scope of the current paper, but you may want to refer to a recent controversy on PTH levels in CKD children reviewing the current evidence on PTH in pediatric CKD : Treatment of hyperphosphatemia: the dangers of high PTH levels. Bacchetta J. Pediatr Nephrol. 2020 Mar;35(3):493-500. doi: 10.1007/s00467-019-04400-w. Epub 2019, versus Treatment of hyperphosphatemia: the dangers of aiming for normal PTH levels. Haffner D, Leifheit-Nestler M. Pediatr Nephrol. 2020 Mar;35(3):485-491. doi: 10.1007/s00467-019-04399-0. Epub 2019 Dec 10...
- In describing the pilot investigation performed by the authors (p4, 1st para) it is briefly mentioned that the bone impairment of cystinosis in the youthful subjects (and a mouse model) impacts upon cortical rather than trabecular bone which of course is usually more vulnerable with its larger surface are for bone cell activity and functional impairment. This seems such a key observation of fundamental consequence (e.g. concerning intramembranous versus endochondral bone behaviour) that to my mind it merits more prominance (maybe even a mention in the Abstract?).
Thanks for this comment. We did not expand on the concept in the discussion since we have very few pathophysiological hypotheses at the moment to explain such an observation. Indeed, even though the seminal paper from Cherqui showed cortical impairment similarly to what is observed in patients, things are not that clear in the paper from Battafarano, showing both cortical and trabecular impairment. As such, we believe that it may be too early to insist on this cortical aspect in the absence of confirmatory data.
- Editorially there is some repetition. For example, the final sentence of the Abstract is repeated verbatim as the final sentence of the Introduction. Also the word "clearly" is overused throughout without justification, as is the meaningless (in this context) word "indeed". Again "dramatically" appears at least twice and has no place at all in good science, and there is no such word as "combinatory" (p4, last para), nor "dysregulation" (p7, bottom line), and "crosstalk" is not in my dictionary. There is also a tendency to overstate by adding adjectives e.g. "very," "considerably" and "paramount importance" (p.8) when paradoxically understatement can be more compelling.
Thank you for this comment. We have modified the manuscript accordingly. However we have kept the word “cross-talk” that has already been published by other teams in the field of bone biology and reflects the interactions between osteoblasts and osteoclasts. We have also kept the word “considerably” in the sentence “as they can considerably affect their quality of life and even require invasive procedures” because we feel that for patients, the quality of life can be really impaired by bone fracture, pains and deformations.
The last sentence of the introduction have been changed to line 70-73 : “The purpose of this review is to report the clinical update on bone pathology in nephropathic cystinosis and provide a summary of basic research findings on bone cell impairment that contribute to unbalanced bone remodeling in this orphan disease and finally to discuss new working hypotheses that deserve future research in the field”.
- The concluding paragraph is carelessly written in comparison with what has gone before and rather than stamping authority on the review it diminishes it, leaving the reader short-changed and dubious. For example, the device "on the one hand" is a useful one but must be partnered with "on the other hand" or it is nonsensical.
Thank you for this comment. We have modified as follows: “Bone impairment in cystinosis is now recognized as a late-onset complication of a multi-systemic disease. It is important that physicians take these potential complications into account in the management of patients with nephropathic cystinosis, as they can considerably affect their quality of life and even require invasive procedures. International guidelines have been recently published to improve the diagnosis and management of CMBD, although the underlying pathophysiology remains to be fully elucidated. In the future, understanding the differential impact of the different CTNS mutations and/or cysteamine therapy on bone cell biology and the molecular mechanisms behind these differential effects should lead to more targeted therapeutic strategies.”.
Round 2
Reviewer 1 Report
The authors addressed all the comments, in particular they completed paragraph 1.2 and expanded the prospective fieldwork with a new paragraph. Therefore, this reviewer believes that the manuscript is now acceptable for publication in IJMS.
Reviewer 2 Report
Much improved and more clearly presents the unknwns in metabolic bone disease of cystinosis
Reviewer 4 Report
Thank you for the thoughtful response. I am satisfied that the authors have attended to the comments made and improved the clarity of the manuscript where requested. The new table is a welcome addition.